# The Expression of IL-1β Correlates with the Expression of Galectin-3 in the Tissue at the Maternal–Fetal Interface during the Term and Preterm Labor

**DOI:** 10.3390/jcm11216521

**Published:** 2022-11-03

**Authors:** Nikola Jovic, Marija Milovanovic, Jovana Joksimovic Jovic, Marija Bicanin Ilic, Dejana Rakic, Vladimir Milenkovic, Bojana Stojanovic, Jelena Milovanovic, Aleksandar Arsenijevic, Nebojsa Arsenijevic, Mirjana Varjacic

**Affiliations:** 1Clinic for Gynecology and Obstetrics, University Clinical Centre of Kragujevac, 34000 Kragujevac, Serbia; 2Department of Gynecology and Obstetrics, Faculty of Medical Sciences, University of Kragujevac, 34000 Kragujevac, Serbia; 3Center for Molecular Medicine and Stem Cell Research, Faculty of Medical Sciences, University of Kragujevac, 34000 Kragujevac, Serbia; 4Department of Physiology Faculty of Medical Sciences, University of Kragujevac, 34000 Kragujevac, Serbia; 5Clinic for Thoracic Surgery, Clinical Center of Serbia, 11000 Belgrade, Serbia; 6Department of Pathophysiology, Faculty of Medical Sciences, University of Kragujevac, 34000 Kragujevac, Serbia; 7Department of Histology and Embriology, Faculty of Medical Sciences, University of Kragujevac, 34000 Kragujevac, Serbia

**Keywords:** Galectin-3, IL-1β, preterm birth, term birth, chorioamnionitis

## Abstract

The inflammatory processes that occur at the maternal–fetal interface are considered one of the factors that are responsible for preterm birth. The pro-inflammatory roles of the Gal-3-induced activation of NLRP3 inflammasome and the consecutive production of IL-1β have been described in several acute and chronic inflammatory diseases, but the role of this inflammatory axis in parturition has not been studied. The aim of this study was to analyze the protein expression of Gal-3, NLRP3, and IL-1β in the decidua, villi, and fetal membranes, and to analyze their mutual correlation and correlation with the clinical parameters of inflammation in preterm birth (PTB) and term birth (TB). The study included 40 women that underwent a preterm birth (gestational age of 25.0–36.6) and histological chorioamnionitis (PTB) and control subjects, 22 women that underwent a term birth (gestational age of 37.0–41.6) without histological chorioamnionitis (TB). An analysis of the tissue sections that were stained with anti- Gal-3, -NLRP3, and -IL-1β antibodies was assessed by three independent investigators. The expression levels of Gal-3 and IL-1β were significantly higher (*p* < 0.001) in the decidua, villi, and fetal membranes in the PTB group when they compared to those of the TB group, while there was no difference in the expression of NLRP3. A further analysis revealed that there was no correlation between the protein expression of NLRP3 and the expression of Gal-3 and IL-1β, but there was a correlation between the expression of Gal-3 and IL-1β in decidua (R = 0.401; *p* = 0.008), villi (R = 0.301; *p* = 0.042) and the fetal membranes (R = 0.428; *p* = 0.002) in both of the groups, PTB and TB. In addition, the expression of Gal-3 and IL-1β in decidua and the fetal membranes was in correlation with the parameters of inflammation in the maternal and fetal blood (C-reactive protein, leukocyte number, and fibrinogen). The strong correlation between the expression of Gal-3 and IL-1β in the placental and fetal tissues during labor indicates that Gal-3 may participate in the regulation of the inflammatory processes in the placenta, leading to increased production of IL-1β, a cytokine that plays the main role in both term and preterm birth.

## 1. Introduction

Preterm birth (PTB) is defined as a delivery before the 37th week of gestation [1], and it is a leading cause of perinatal mortality [2,3]. Also, long-term infant morbidity including cerebral palsy, bronchopulmonary dysplasia, an intraventricular hemorrhage, and necrotizing enterocolitis are in more than 50% of the cases associated with PTB [4,5,6]. A spontaneous PTB which accounts for 25% or 45% of the PTB cases can be caused by a vascular disease, and uterine over-distention [1], but the leading cause of spontaneous PTB is infection and infection-induced inflammation [7,8]. However, intra-amniotic inflammation without detectable microorganisms has been shown in a subset of women with PTB [9,10]. Inflammatory reactions that take place in the fetal membranes (amnion and chorion) and are expanded to the placenta and decidua are defined as chorioamnionitis [11]. The estimated frequency of chorioamnionitis is 2–4% in term births [12,13] and 40–70% in preterm births [12,14]. Chorioamnionitis is mostly associated with microorganisms, but it also can occur as a sterile intra-amniotic inflammation that is initiated by endogenous danger signals that are derived from damaged and necrotic cells, which are termed damage-associated molecular patterns (DAMPs) or alarmins, environmental pollutants, cigarette smoke, and similar toxins [15,16,17,18,19].

Both of these conditions, the microbial-associated and sterile intra-amniotic inflammation types, involve the release of pro-inflammatory cytokine, interleukin (IL)-1β [20]. The role of IL1-β in the process of parturition, both at term and preterm, is mostly known. IL-1β together with TNF-α (Tumor necrosis factor-α) stimulates the biosynthesis of prostaglandins from arachidonic acid, primarily prostaglandin E2 (Prostaglandin E2—PGE2) with/without prostaglandin F2α (Prostaglandin F2α—PGF2α) by human chorionic, amnion and decidual cells, and through increasing cyclooxygenase activity (Cyclooxygenase 2—COX2) [21,22]. These two prostaglandins are known to lead to cervical ripening and uterine contractions that precede parturition [23]. The expression of IL-1β has been demonstrated in the myometrium, cervix, and amniotic membranes immediately and during labor regardless of the presence/absence of infection [24,25]. Its values increase in the amniotic fluid during preterm delivery, especially if it is induced by infection [26,27]. It is also detected in the mother’s blood during preterm delivery [28], in the placenta and fetal membranes during infection, but also in the tissue of the uterus [29,30,31,32]. IL-1 is also considered as a key driver of the inflammation in preterm labor [23]. The concentration of IL-1 in the amniotic fluid is significantly higher in preterm births than it is in term births.

Galectin-3 (Gal-3) is an endogenous lectin with various immunoregulatory effects, playing both disease-exacerbating and protective roles in inflammatory diseases [33,34,35,36,37,38,39]. Gal-3 participates in several ways in the innate immune response against invading pathogens. Galectin-3 has been proposed to function not only as a pattern-recognition receptor (PRR) that binds lipopolysaccharide (LPS), performing pro-inflammatory actions by promoting the infiltration of immune cells to the infected sites [40], but it also can be released as a DAMP [41]. The opposite roles of Gal-3 interaction with the pathogen-associated molecular pattern molecules (PAMPs) have been reported, marking Gal-3 as a negative regulator of LPS-induced inflammation [42], but also as the main amplifier of LPS-induced IL-1β production and monocyte chemotaxis [43,44]. NLRP3 (NOD-, LRR- and pyrin domain-containing protein 3) is an intracellular molecule that detects a broad range of microbial motifs, endogenous danger signals, PAMPs, and environmental irritants, resulting in the formation and activation of the NLRP3 inflammasome [45]. NLRP3 inflammasome activation results in the caspase 1-dependent release of the pro-inflammatory cytokine IL-1β [46], a cytokine that plays the central role in the process of labor [23]. Gal-3 has been described as one of the activators of NLRP3 and the consequential triggering of the IL-17 pro-inflammatory cascade [45]. The pro-inflammatory roles of the Gal-3-induced activation of NLRP3 inflammasome and the production of IL-1β have been described in several acute and chronic inflammatory diseases [35,36,39,47].

Although local inflammation at the maternal–fetal interface is much more relevant for pregnancy and the process of parturition [48], previous studies have mainly concentrated on exploring the immune and inflammatory responses in the peripheral blood to reveal the mechanisms that are responsible for pathogenic pregnancy [49,50]. The expression of Gal-3 was detected in maternal decidual cells, the mRNA, and at the protein level [51], as well as in trophoblast lineage cells [52] where it is associated with the differentiation of the cytotrophoblasts [53,54], the trophoblast invasion machinery [55], and syncytialization [56]. Studies in mice revealed the role of Gal-3 in successful implantation [56] and symmetric placental growth [57]. However, the expression of Gal-3 and NLRP3, molecules with important roles in the regulation of inflammatory processes, in placental tissue in the context of parturition were studied in only a few studies with varying results.

In this study, we have analyzed the protein expression of Gal-3, NLRP3, and IL-1β in the decidua, villi, and fetal membranes during term labor and PTB, their mutual correlation, and their correlation with the clinical parameters of inflammation to explore their eventual role in the inflammatory processes that take place during parturition.

## 2. Materials and Methods

### 2.1. Ethics Statement

The subjects for the study were selected from women that were experiencing childbirth at the Obstetrics and Gynaecology Department of Clinical Centre Kragujevac. The study was conducted in accordance with the Declaration of Helsinki and the Ethics Committee of the Clinical Centre “Kragujevac”, whereby written consent was obtained from each patient that was included in the study. Approval for this study was obtained from the Ethics Committee of the Faculty of Medical Sciences University of Kragujevac, number 01-12994 and the Ethics Committee of the Clinical Center Kragujevac, number 01/17-4259.

### 2.2. Study Population

The study included 40 women that underwent a preterm birth (gestational age of 25.0–36.6) and histological chorioamnionitis (PTB) and control subjects, 22 women that underwent a term birth (gestational age of 37.0–41.6) without histological chorioamnionitis (TB). Histologic chorioamnionitis was defined as the presence of acute inflammatory changes during the examination of the membrane role of the placenta. Samples that were used in the study were tissues of fetal membranes and placental discs. The criteria for selecting subjects for the study were as follows: women with no comorbidities, specifically gestation-associated diseases (chronic or gestational diabetes, congenital or acquired thrombophilia, hypertension, chronic inflammatory, and autoimmune diseases) and without having undergone antibiotic and corticosteroid therapy for at least two weeks before parturition. In addition, all of the women had regular dental visits during their pregnancy, and no dental infections were registered in any of the patients. In this study, all of the patients underwent vaginal deliveries, without oxytocin and prostaglandin induction.

### 2.3. Evaluation of Biochemical Parameters in Sera and Blood Cell Number

Serum levels of C-reactive protein, fibrinogen, and leukocyte number in the blood, which were obtained within the first minute after cord clamping, were routinely determined by the standard methods that were suggested by the IFCC (International Federation for Clinical Chemistry and Laboratory Medicine) in the Central Biochemical Laboratory of the Clinical Centre Kragujevac, Serbia.

### 2.4. Immunohistochemistry

Paraffin-embedded samples were consecutively cut to a thickness of 4–5 μm and placed on SuperFrost plus slides (Thermo Scientific, Waltham, MA, USA). Each section was deparaffinized and rehydrated with graded ethanol. Antigen retrieval was performed by microwave heating for 20 min in 10 mM sodium citrate buffer (pH 6.0). The activity of endogenous peroxidase was blocked with a 3% hydrogen peroxide solution (Abcam, Cambridge, UK) for 10 min at room temperature. After washing them with PBS, the slides were incubated with primary mono/polyclonal antibodies against NLRP3 (ab214185, Abcam, Cambridge, UK, at a 1:500 dilution), IL-1β (ab9722, Abcam, Cambridge, UK, at 1 μg/mL), and galectin-3 (ab53082, Abcam, Cambridge, UK, at a 1:100 dilution) overnight in a humid chamber at 4 °C. Antibody-treated sections were washed and stained using rabbit-specific HRP/AEC and HRP/DAB kits (Abcam, Cambridge, UK) according to the manufacturer’s recommendations. For negative controls, identical protocol for immunohistochemistry was performed, but primary antibody was omitted (Figure 1). All of the sections were counterstained in Mayer’s hematoxylin, dehydrated with alcohol, and mounted. An Olympus microscope (BX50 model) equipped with a digital camera was used to prepare microphotographs with magnifications of 100× or 400×.

### 2.5. Immunohistochemistry Scoring

Staining was assessed by three independent investigators for the staining intensity, and the percentages of the cells that were stained were decided. The staining intensity samples were scored as follows: 0 for no reactivity, 1 for the presence of weakly stained ones, 2 for moderately stained ones, and 3 for strongly stained cells. Representative sections of scores 1, 2, and 3 for Gal-3 in decidua, villi and fetal membranes tissues are presented on Figure 2. The percent of stained cells was scored as follows: 0 for no positive cells, 1 for infrequent and isolated positive cells, 2 for <50% cells positive, and 3 for >51%, positive cells. The two scores were multiplied to give a final score that is between 0 and 9 (35). Samples were considered positive when the final score was >2. Slides were analyzed using Olympus BX51 microscope, and digital images were acquired by using Olympus digital camera (DP71).

### 2.6. Statistical Analysis

The data were analyzed using the commercially available SPSS 20.0 software (IBM corporation, Armonk, NY, USA). The results were reported as the mean and standard error of the mean (SEM). Results were analyzed using the Student’s *t*-test for independent samples if the data had normal distribution or Mann–Whitney U test for data without normal distribution. Spearman’s correlation assessed the possible relationship between the selected variables. The strength of correlation was defined as negative or positive weak (−0.3 to −0.1 or 0.1 to 0.3), moderate (−0.5 to −0.3 or 0.3 to 0.5), or strong (−1.0 to −0.5 or 1.0 to 0.5). Statistical significance was set at *p* < 0.05.

## 3. Results

### 3.1. Maternal and Fetal Inflammatory Blood Parameters Are Increased in PTB Subject

Forty-four PTB subjects and twenty-two TB subjects were enrolled in the study. The clinical characteristics of these patients are presented in Table 1. There was no significant difference between the groups in their age, i.e., there was a mean age of 29.68 ± 7.19 in the PTB group versus 32.55 ± 6.22 in the TB group. Additionally, there was no difference in the number of previous labors, i.e., there were 1.77 ± 0.75 in the preterm birth group versus 1.65 ± 0.74 in the TB group. Using the criteria for the selection of the subjects for PTB and TB groups, a significant difference was found in the gestational ages 30.20 ± 3.84 in the PTB group and 38.95 ± 1.53 in TB group (*p* < 0.01).

The serum levels of C-reactive protein and fibrinogen and the leukocyte number were analyzed in the maternal and fetal blood in both of the defined groups. The mean concentrations of the C-reactive protein and fibrinogen were significantly higher in the maternal and fetal blood in PTB subjects when they were compared to those of the TB group: mean ± standard error 44.74 ± 44.89 versus 7.34 ± 3.36  mg/L, *p* < 0.01 for maternal C-reactive protein; 29.89 ± 30.68 versus 6.09 ± 2.96  mg/L, *p* < 0.01 for fetal C-reactive protein; 5.70 ± 1.78 versus 4.22 ± 0.83  mg/L, *p* < 0.01 for maternal fibrinogen; 4.88 ± 1.44 versus 3.97 ± 0.74  mg/L, *p* < 0.01 for fetal fibrinogen. The absolute number of leukocytes was also significantly (*p* < 0.01) higher in the maternal (18.46 ± 4.47 × 10^9^/L) and fetal blood (15.80 ± 3.95 × 10^9^/L) in the PTB group when they were compared to the numbers of these cells in the maternal (9.89 ± 3.04 × 10^9^/L) and fetal (8.53 ± 2.29 × 10^9^/L) blood in the TB group (Table 2).

### 3.2. Expression Levels of Galectin-3 and IL-1β and Percentages of Positive Fetal Membranes and Placental Disc Samples Are Higher in the PTB Group

The percentage of the samples that express the analyzed molecules is presented in Figure 3. A low percentage of samples expressing Gal-3 in the villi, NLRP3 in the fetal membranes, and IL-1β in the decidua and villi were detected in the TB group. Contrary to this, these molecules were expressed in a high percentage in the PTB group. A higher percentage of the positive tested samples was found in the PTB group versus that in the TB group for Gal-3 (97.72% in decidua in PTB group versus 81.81% in TB; 75.00% in villi in PTB versus 4.54% in TB; and 88.63% in fetal membranes in PTB versus 31.81% in TB group) (Figure 1a), NLRP3 (97.72% in decidua in PTB versus 95.45% in TB; 61.36% in villi in PTB versus 63.63% in TB; and 88.63% in fetal membranes in PTB versus 13.63% in TB) (Figure 3b), and IL-1β (77.27% in decidua in PTB versus 9.09% in TB; 18.18% in villi in PTB versus 4.54% in TB; and 77.27% in fetal membranes in PTB versus 45.45% in TB) (Figure 3c).

The expression levels of Gal-3 and IL-1β were significantly higher (*p* < 0.001) in the decidua, villi, and fetal membranes in the PTB group when they were compared to those of the TB group (Figure 4). Higher expression levels of Gal-3 and IL-1β were detected in the decidua and fetal membranes than in the villi in both of the groups, PTB and TB (Figure 5). There was no significant difference in the expression of NLRP3 between PTB and TB in all of the examined tissues placentas (decidua and villi) and in the fetal membranes (Figure 5).

The analysis of the localization of the Gal-3, IL-1β, and NLRP3 staining revealed similar patterns in both the PTB and term birth samples with the only difference being in the staining intensity, hence representative sections from the PTB group are presented in Figure 5. NLRP3 expression was found in the cytoplasm of the decidual and amniotic cells (Figure 5). In chorionic villi, a very weak NLRP3 expression was detected in the cytoplasm of the trophoblast cells, while a stronger expression of it was detected in the intravillous endothelial cells (Figure 5). Gal-3 was expressed in the cytoplasm and also some nuclei of the decidual and amniotic cells. However, in the chorionic villi, Gal-3 was detected in the cytoplasm of the trophoblast cells and the mononuclear inflammatory cells and also extracellularly in the villous connective tissue (Figure 5). IL-1β was detected in the cytoplasm of the decidual and amniotic cells, while in the villi, IL-1β was mostly detected extracellularly in the intravillous connective tissue, which is similar to the pattern of Gal-3 expression, and also in the extravillous space (Figure 5). In addition to the extracellular localization, IL-1β was detected in the cytoplasm of the trophoblast cells and also in the mononuclear inflammatory cells in the chorionic villi (Figure 5). Additionally, IL-1β was detected in the mononuclear inflammatory cells in chorion and also in the extracellular space around these inflammatory cells (Figure 5).

A further analysis revealed that there was no correlation between the protein expressions of NLRP3 and Gal-3, and also between those of NLRP3 and IL-1β (data are not shown).

Interestingly there was significant correlation between the expressions of Gal-3 and IL-1β in the decidua (R = 0.401; *p* = 0.008), villi (R = 0.301; *p* = 0.042), and fetal membranes (R = 0.428; *p* = 0.002), as shown in Figure 6b.

### 3.3. Correlation between Placental and Fetal Membranes Expression of Gal-3 and IL-1β and Inflammation

To test the hypothesis that the protein expressions of NLRP3, Gal-3, and IL-β in the placental tissue and fetal membranes are significantly associated with the parameters of inflammation in the maternal and fetal blood, a statistical analysis of the correlation was performed. As shown in Figure 7, Gal-3 expression in the fetal membranes was associated with the maternal leukocyte number (R = 0.494; *p* = 0.000), CRP (R = 0.273; *p* = 0.05), and fibrinogen (R = 0.354; *p* = 0.010), and the fetal leukocyte number (R = 0.485; *p* = 0.000).

The Gal-3 expression in the decidua was associated with the maternal leukocyte number (R = 0.403; *p* = 0.007) and fibrinogen (R = 0.324; *p* = 0.032) and the fetal blood leukocyte number (R = 0.387; *p* = 0.009) (Figure 8).

The IL-1β expression in the decidua was in correlation with the maternal and fetal leukocyte number (maternal blood R = 0.557; *p* = 0.00; fetal blood R = 0.458; *p* = 0.001), CRP (maternal blood R = 0.451; *p* = 0,001; fetal blood R = 0.403; *p* = 0.003), and fibrinogen (maternal blood R = 0.357; *p* = 0.009; fetal blood R = 0.344; *p* = 0.012), as shown in Figure 9. There was no correlation between the clinical parameters and expression levels of IL-1β in the fetal membranes and NLRP3 in all of the examined tissues.

## 4. Discussion

In this study, we have shown that there are significantly higher expression levels of Gal-3 and IL-1β in the decidua, fetal membranes, and villi in the PTB group with chorioamnionitis in comparison with the TB group without chorioamnionitis. The higher expression of Gal-3 and IL-1β in these tissues in the PTB group with chorioamnionitis was accompanied by the higher serum levels of the markers of inflammation, fibrinogen, CRP, and the leukocyte number in the maternal and fetal blood at delivery. Additionally, we have provided the first evidence that the expression of IL-1β in the decidua and fetal membranes at the time of delivery is in correlation with the expression of Gal-3 in both of the groups, PTB and TB.

Inflammatory processes in different gestational tissues are involved in term and preterm birth [57]. The mediator that is intrinsically involved in the tissue-level transformation of the maternal and fetal intrauterine tissues for labor is IL-1β [58]. IL-1β has been reported as a key inducer of inflammation in PTB by binding to its ubiquitously expressed receptor IL-1RI, which leads to the activation and amplification of the inflammatory cascade [59]. The expression of IL-1β is increased in the decidua, cervix, and fetal membranes during term labor [24,25,60]. However, IL-1β was also reported as the first cytokine that plays a role in preterm labor that is associated with intra-amniotic infections [26,61], and the cytokine that has been administered to mice systemically [62] and intra-amniotic to Rhesus Monkeys [63] induced chorioamnionitis and preterm labor. An increased IL-1β presence in the vaginal washings of women at risk of undergoing a preterm birth has been observed [64]. Additionally, a significant increase in the expression of IL-1β has been reported in the decidua of spontaneous and recurrent miscarriage placentas [65]. Here, we reported a significantly higher expression of IL-1β in the tissues of decidua, villi, and fetal membranes in the PTB group when they were compared to those of the term birth group (Figure 4). Additionally, we found that there was a higher percentage of decidua, villi, and fetal membranes tissue samples which tested positive for IL-1β expression in the PTB group versus to the term birth group (Figure 3). Our results are in line with previous reports regarding the role of IL-1β in the induction and amplification of the inflammatory cascade that leads to preterm delivery [59].

The active form of IL-1β arises from the immature molecule by proteolytic activity of the enzyme caspase-1. The activity of caspase-1 is regulated by the activated NLRP3 inflammasome [66]. A previous study reported an increased NLRP3 expression in the chorioamnion membranes in PTB, which was accompanied by an increased inflammasome activation and an increased expression of the mature form of IL-1β in spontaneous preterm labor with chorioamnionitis [67]. Further, the production of IL-1β in fetal membranes in response to bacterial products is mediated by NLRP3 [68]. Additionally, several experimental studies have reported an important role of NLRP3 inflammasome activation in preterm birth induction in the context of intrauterine inflammation [69,70,71], but a recent study reported the involvement of NLRP3 inflammasome in uterine activation for term labor onset also in human and mouse models [72]. Here, we reported that almost all of the samples of the decidua expressed NLRP3 in both of the groups, with half of villi samples also expressing NLRP3 in both of the groups, while there was a higher percentage of fetal membrane samples in the PTB group that expressed NLRP3 when they were compared to the samples of the term birth group without chorioamnionitis (Figure 3). However, there was no difference in the expression level of NLRP3 between the two groups (Figure 4). Unlike the findings that were reported in previous studies, we have not found a correlation between the expression levels of IL-1β and NLRP3 in the examined tissues.

Besides the inflammasome-mediated production of IL-1β, there are several cell-specific inflammasome-independent processes that are involved in the activation of IL-1β [73]. Increased IL-1β production without the concomitant and spatially coincident increased expression of NLRP3 has been shown in active ulcerous colitis, suggesting that during normal gut homeostasis, the production of IL-1β is mediated by inflammasome-dependent caspase-1, while during active ulcerous colitis, the cleavage of IL-1β is mediated by serine proteases from neutrophils [74]. There is evidence that suggests that innate immune cells (neutrophils, macrophages and mast cells) mediate the process of labor by releasing the proinflammatory factors such as cytokines, chemokines, and matrix metalloproteinases [75]. Inflammatory neutrophils are present in the uterus, decidua, cervix, and fetal membranes during labor [24,76,77] where they participate in the labor by inducing the secretion of inflammatory cytokines and matrix metalloproteinases [78,79]. The inflammatory mediators that are released from the neutrophils participate also in the process of preterm labor [80]. Our results indicate that the maturation of IL-1β in the decidua and fetal membranes during both preterm and term labor could be mostly mediated by the enzymes that are released from the neutrophils or other innate immune cells which are known to infiltrate these tissues during labor [75].

The production of IL-1β can be stimulated [81] and amplified by Gal-3 [82]. In accordance with this, we found a significant correlation between the expression of Gal-3 and IL-1β in the decidua, villi, and fetal membranes (Figure 6) and a similar pattern of Gal-3 and IL-1β localization in the chorionic villi (Figure 5). A previous study reported an increased expression of Gal-3 at mRNA and protein levels in human decidua at term with spontaneous labor [83]. A lower expression of Gal-3 at both of the levels of mRNA and protein was found in the decidua in spontaneous labor at term when they were compared to those of the samples from patients undergoing an elective Cesarean section at term [83]. Another study reported an increased Gal-3 expression in the fetal membranes and in the amniotic epithelium in chorioamniotic infection and the preterm premature rupture of the membranes, suggesting the role of Gal-3 in the inflammatory responses in chorioamnionitis and/or in direct interaction with the pathogens [57]. Additionally, elevated levels of Gal-3 in the maternal serum in pregnancies that are complicated by the preterm pre-labor rupture of the membranes [84] and the blood of preterm infants born in an inflammatory milieu has been reported, but there are no data as to whether Gal-3 mediates the inflammation-induced preterm birth, and this could, therefore, be a target for clinical trials [85]. A recent study reported an increased expression and strong staining results in placental extravillous trophoblast tissues, and also higher levels of Gal-3 in the maternal blood in spontaneous preterm births when they were compared to those of the spontaneous term pregnancies during labor [86]. An increased expression was associated with a higher percentage of trophoblasts with short telomeres, indicating that increased senescence and inflammation might be factors that are involved in spontaneous preterm labor [86]. Our results of the increased expression of Gal-3 in the tissue samples of decidua, villi, and fetal membranes in the cases of PTB with chorioamnionitis (Figure 4 and Figure 6) are in line with the results of previous studies. In correlation with this are also results of an experimental study on mice that reported a significant upregulation of Gal-3 in the placenta, amniotic fluid, and serum of the mice that underwent dental infection-induced preterm births [87]. Gal-3 also stimulated the production of inflammatory cytokines that contribute to the development of PTB, suggesting the immunomodulatory/proinflammatory role of Gal-3 in PTB and indicating Gal-3 as a potential therapy and/or diagnostic target that may reduce the occurrence of PTB [87].

The strength of our study were our rigorous exclusion criteria. Further, the pregnant women for whom, after giving birth, were subject to a detailed medical examination, which revealed the presence of criteria that were a priori exclusionary, were subsequently excluded from the study, which additionally affected the total number of subjects in it. The limitation of our study is its sample size. The frequency of diseases accompanying pregnancy is increasing in modern perinatology, and the percentage of surgical terminations of pregnancy is increasing also. It is very difficult to perform a “uniform” study of preterm birth, which is managed naturally without induction, which occurs quickly so that the pregnant woman does not receive a treatment for fetal lung maturation and does not undergo an antibiotic therapy for at least two weeks. Moreover, during pregnancy, such a patient should not have any acute or chronic, inflammatory, systemic, or autoimmune diseases, which should be confirmed by medical examinations and documentation. The mentioned factors explain the reason for the number of respondents in our study.

## 5. Conclusions

After calculating the correlation between the expression levels of IL-1β and Gal-3 in the decidua and fetal membranes and the correlation between the Gal-3 expression and serum levels of the parameters of inflammation and the leukocyte numbers in the blood in the PTB and term birth groups, it was found in this study that that Gal-3 has a pro-inflammatory role during labor. There being higher systemic parameters of inflammation which are accompanied by higher expressions of Gal-3 and IL-1β in the decidua and fetal membranes in the PTB group versus the term delivery group also suggest that Gal-3 plays a role in the amplification of inflammation that leads to preterm labor. These results indicate that Gal-3 may be further explored as a potential therapeutic target in the prevention of preterm labor.

## Figures and Tables

**Figure 1 jcm-11-06521-f001:**
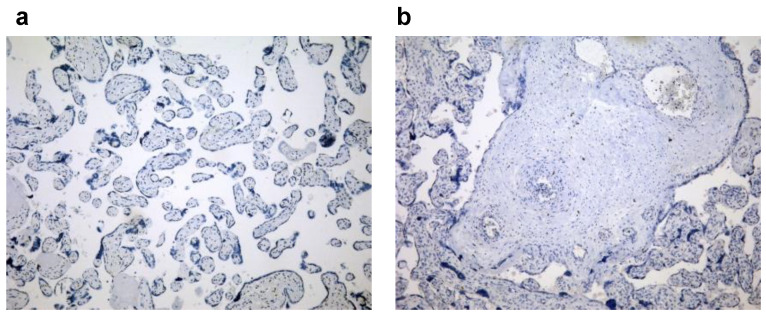
Sections of negative staining controls of placental tissue (magnification 100×), (**a**) HRP/DAB, (**b**) AEC/DAB kit.

**Figure 2 jcm-11-06521-f002:**
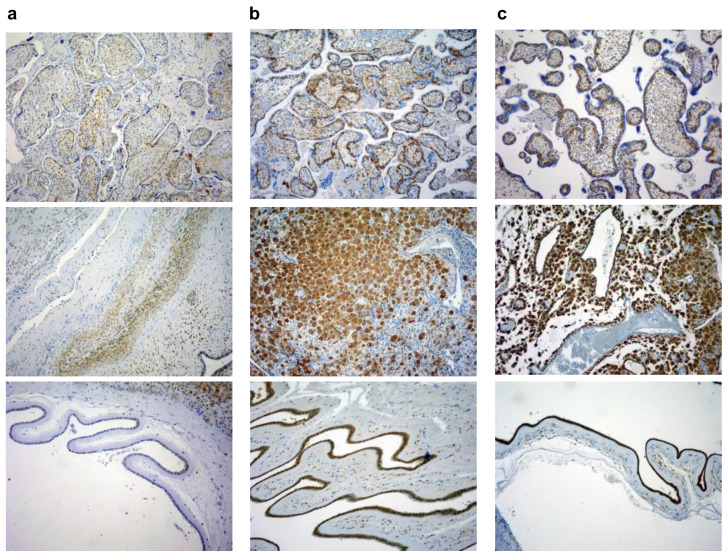
Example of staining intensity scores 1 (**a**), 2 (**b**), and 3 (**c**) for Gal-3 in decidua, villi and fetal membranes tissues (magnification 100×).

**Figure 3 jcm-11-06521-f003:**
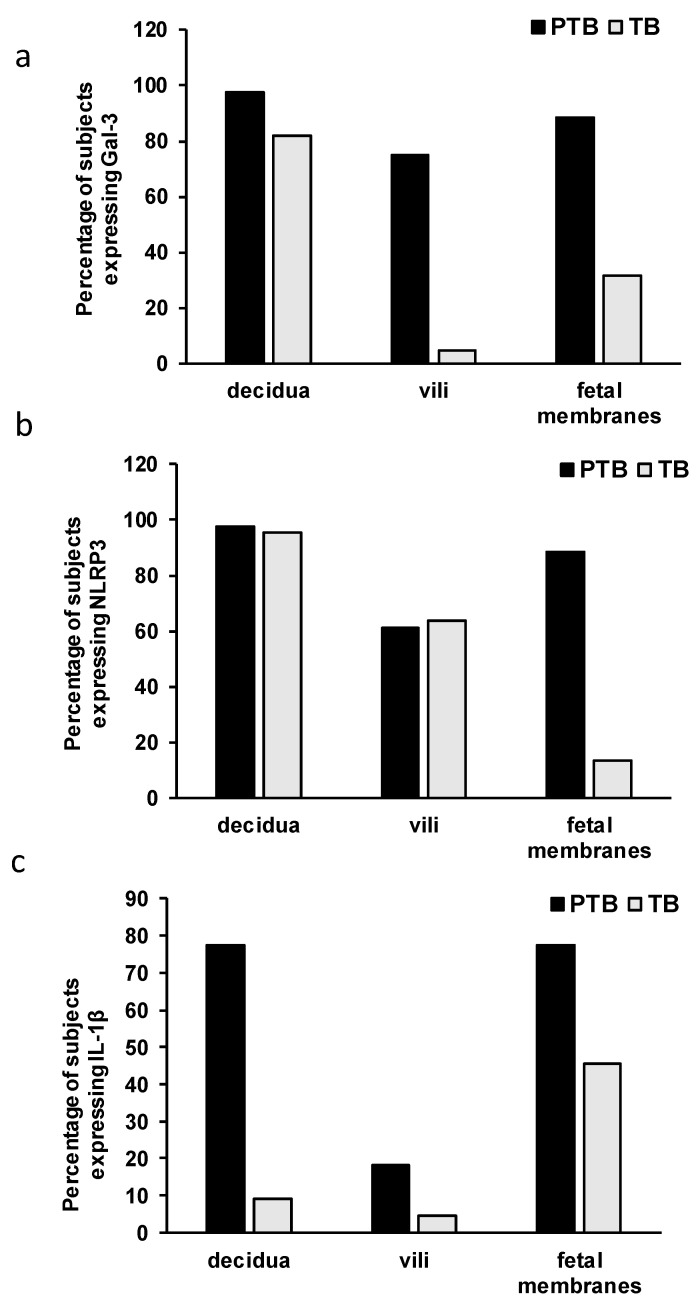
Incidence of Gal-3 (**a**), NLRP3 (**b**), and IL-1β (**c**) in tissues of decidua, villi, and fetal membranes obtained during preterm (PTB) and term (TB) births.

**Figure 4 jcm-11-06521-f004:**
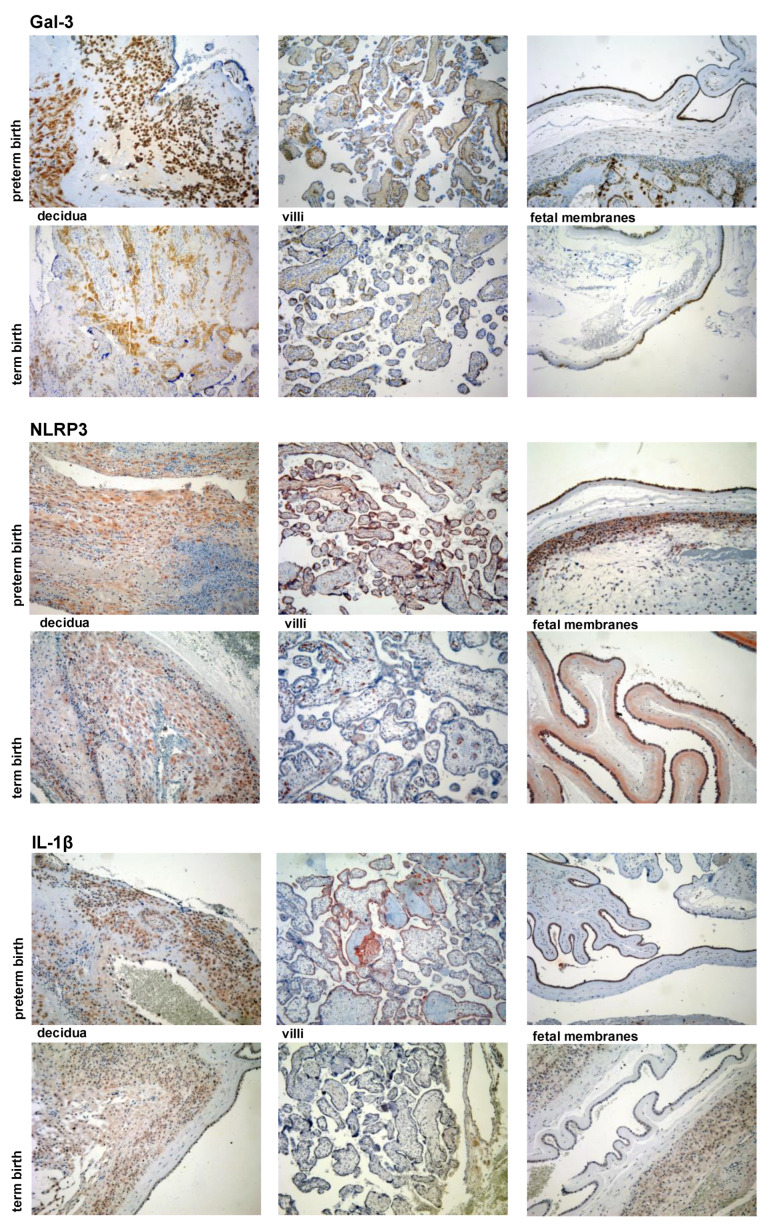
Representative sections of Gal-3, IL-1β, and NLRP3 immunohistochemistry of decidua, villi and fetal membranes tissues obtained during preterm and term births (magnification 100×).

**Figure 5 jcm-11-06521-f005:**
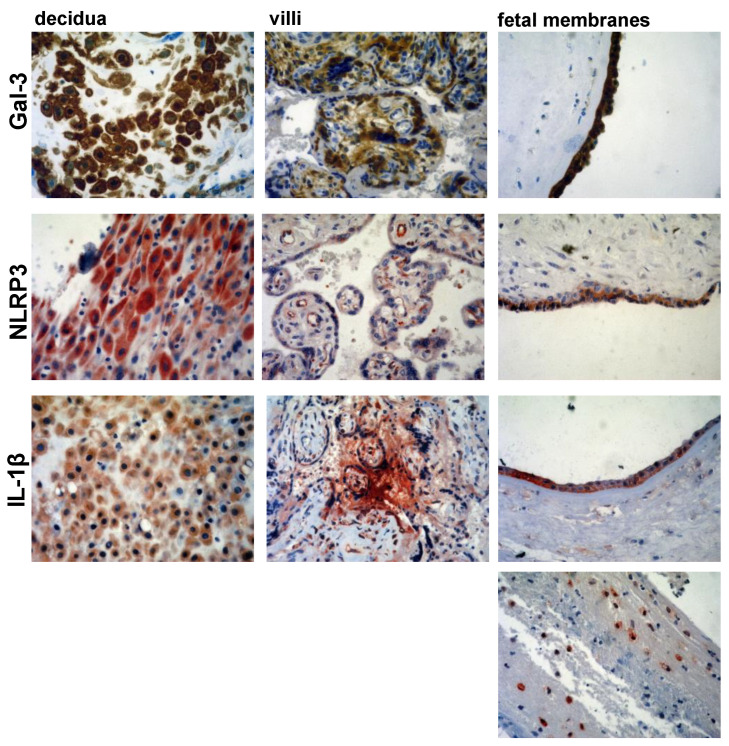
Gal-3, IL-1β, and NLRP3 staining of decidual, amniotic, and trophoblast cells in decidua, villi, and fetal membranes tissues which were obtained during preterm births (magnification 400×).

**Figure 6 jcm-11-06521-f006:**
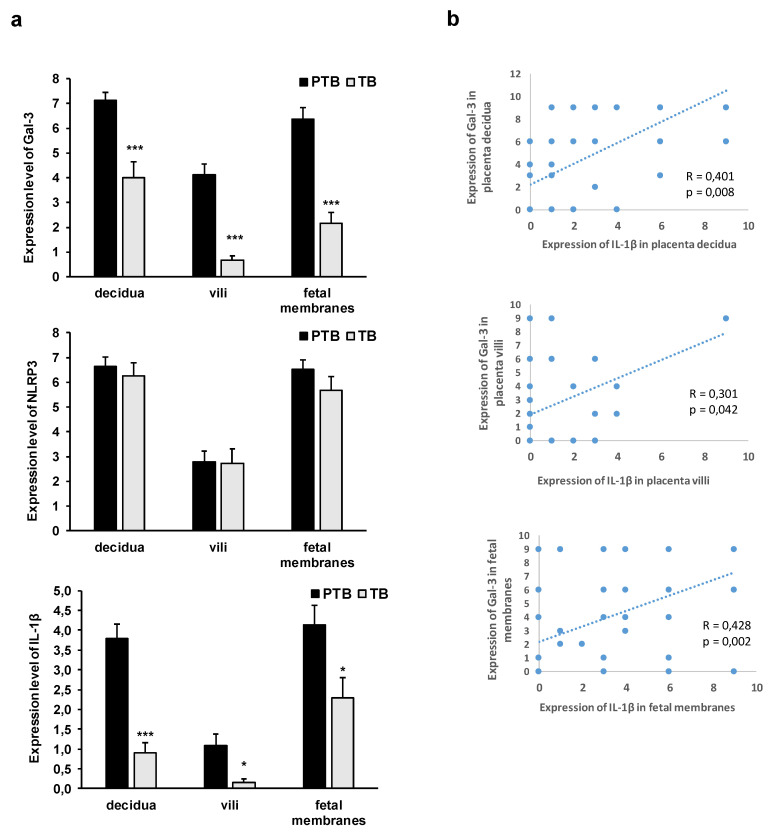
Level/intensity of Gal-3, IL-1β, and NLRP3 protein expression in tissues of decidua, villi, and fetal membranes obtained during preterm (PTB) and term (TB) births, semi-quantification (**a**) and correlation in expression of Gal-3 and IL-1β in decidua, villi and fetal membranes (**b**)*. **—significance level at <0.05; ***—significance level at *p* < 0.001.

**Figure 7 jcm-11-06521-f007:**
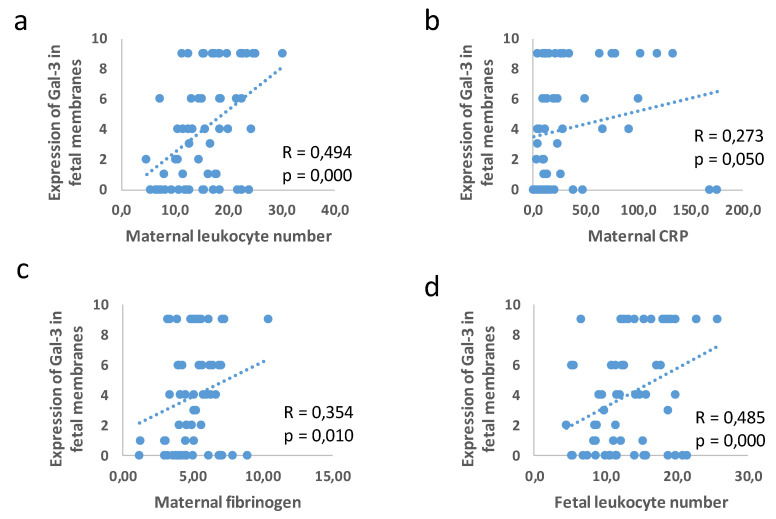
Correlation in expression of Gal-3 in fetal membranes and leukocyte number in maternal blood (**a**), maternal CRP (**b**), maternal fibrinogen (**c**), and leukocyte number in fetal blood (**d**).

**Figure 8 jcm-11-06521-f008:**
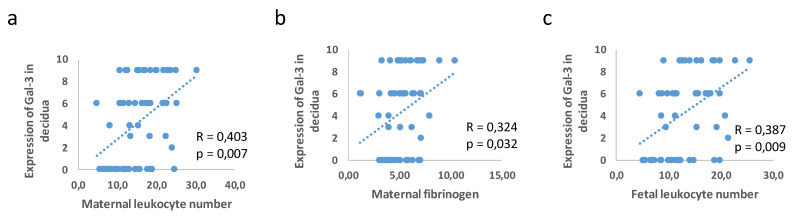
Correlation in expression of Gal-3 in decidua and leukocyte number in maternal blood (**a**), maternal fibrinogen (**b**), and leukocyte number in fetal blood (**c**).

**Figure 9 jcm-11-06521-f009:**
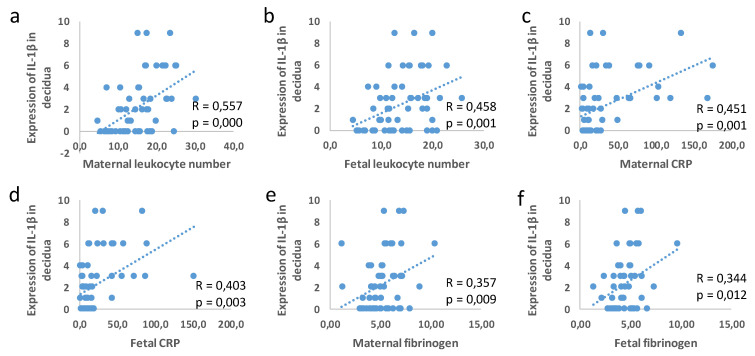
Correlation in expression of IL-1β in decidua and leukocyte number in maternal blood (**a**), leukocyte number in fetal blood (**b**), maternal levels of CRP (**c**), fetal levels of CRP (**d**), maternal fibrinogen (**e**), fetal levels of fibrinogen (**f**).

**Table 1 jcm-11-06521-t001:** General characteristics of the patients.

General Characteristics of the Patients	Control Group (Term Delivery) (n = 22)	Experimental Group (Preterm Delivery) (n = 40)
Age	32.55 ± 6.22	29.68 ± 7.19
Gestational age	38.95 ± 1.53	30.20 ± 3.84 **

Values are expressed as mean (Mean) ± standard deviation (SD). ** denotes statistical significance compared to term delivery (*p* < 0.01).

**Table 2 jcm-11-06521-t002:** Inflammation parameters which were determined from maternal systemic circulation and umbilical cord blood.

Parameters of Inflammation		Control Group (Term Delivery) (n = 22)	Experimental Group (Preterm Delivery) (n = 40)
From maternal blood	Leukocytes (×10^9^/L)	9.89 ± 3.04	18.46 ± 4.47 **
	CRP (mg/L)	7.34 ± 3.36	44.74 ± 44.89 **
	Fibrinogen (g/L)	4.22 ± 0.83	5.70 ± 1.78 **
From fetal blood	Leukocytes (×10^9^/L)	8.53 ± 2.29	15.80 ± 3.95 **
	CRP (mg/L)	6.09 ± 2.96	29.89 ± 30.68 **
	Fibrinogen (g/L)	3.97 ± 0.74	4.88 ± 1.44 **

CRP—C-reactive protein. Values are expressed as mean (Mean) ± standard deviation (SD). ** denotes statistical significance compared to term delivery (*p* < 0.01).

## Data Availability

The data presented in this study are available on request from the corresponding author.

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
