# Peer review of "The Expression of IL-1β Correlates with the Expression of Galectin-3 in the Tissue at the Maternal–Fetal Interface during the Term and Preterm Labor"

_jcm, 2022, doi:10.3390/jcm11216521_

Round 1
Reviewer 1 Report
Dear authors,
thanks for prvoding a new and revised version of your manuscript. The graphs of the immunohistochemistry have improved quality now and the changes in the manuscript seem to be appropriate. Looking forward to further clinical studies. Best regards
Author Response
Thank you very much for your comments.
Reviewer 2 Report
Authors presented an interesting study about to of Gal-3, NLRP3, and IL-1β role in inflammatory processes during parturition. However, they should improve the manuscript as detailed below: The text should be revised by a native speaker to remove several typos. It would be desirable to increase the number of cases since it is difficult to draw conclusions with such a small number of patients. Moreover, authors should specify how simple size was calculated. Finally, the discussion should be improved citing relevant and novel key articles about the topic.
Author Response
Thank you very much for your comments.
- The text was revised by English professional.
- It is very difficult to get a ``uniform'' preterm delivery, managed naturally without induction, which occurred so quickly that the pregnant woman did not receive therapy for fetal pulmonary maturation, and that she had not been receiving antibiotic therapy for at least two weeks. The criteria for inclusion in the study were also absence of any acute or chronic systemic inflammatory, or autoimmune disease during the pregnancy, known or confirmed by medical tests or documentation. These factors explain the number participants in our study. Further, if any condition that was a priori exclusionary was revealed after delivery, by a detailed medical examination or investigation, such respondents were subsequently excluded from the study, which additionally affected the total number of participants. Also, considering the time limit of the study, rigorous exclusion criteria, and considering the expiration date of the chemicals obtained for the experiments we did not have enough time to form the larger groups.
- Study sample was calculated based on the assumption that requires the largest sample, i.e. the expected smallest difference in the investigated parameters (IL-1β expression in placenta I studies of similar design) between the experimental and control groups, for the level of significance α=0,05 and statistical power ≥ 80%.
- We discussed several novel article related to the topic and we believe that the manuscript is improved now.
Reviewer 3 Report
It would be interesting to know if the evaluation of biochemical parameter has been done at the time of delivery or in the prodomical phase and why they haven´t chosen preterm birth without histological chroriamnionitis or term birth with histological chroriamnionitis.
I need to justify the reason why the sample size is different in both groups, I would like than the group of women with term birth has, at least, the same size.
Author Response
The blood for the evaluation of biochemical parameters was obtained within the first minute after cord clamping, and it is added in the revised manuscript (line 132 in revised version).
The frequency of diseases accompanying pregnancy is constantly increasing in modern times perinatology, as well as the percentage of surgical terminations of pregnancy. It is very difficult to get a ``uniform'' preterm delivery, managed naturally without induction, which occurred so quickly that the pregnant woman did not receive therapy for fetal pulmonary maturation, and that she had not been receiving antibiotic therapy for at least two weeks. The criteria for inclusion in the study were also absence of any acute or chronic systemic inflammatory, or autoimmune disease during the pregnancy, known or confirmed by medical tests or documentation. These factors explain the number participants in our study. Further, if any condition that was a priori exclusionary was revealed after delivery, by a detailed medical examination or investigation, such respondents were subsequently excluded from the study, which additionally affected the total number of participants. The basic criterion for division into groups was related to the time of childbirth, that is, whether it is a preterm delivery (experimental group) or a term delivery childbirth (control group). The study was further carried out according to the principle of a double-blind study, we were waiting for the results of rejection/confirmation of chorioamnionitis (in the population of the included participants) 5 to 7 days. The results of the pathohistological analysis confirmed without exception chorioamnionitis in all respondents included in experimental (preterm birth) group, while chorioamnionitis was not found in any case of term in term births.
Considering the time limit of the study, rigorous exclusion criteria, and considering the expiration date of the chemicals obtained for the experiments, we did not have enough time to form the groups of preterm births without chorioamnionitis and term births with chorioamnionitis. However, we included all cases that fitted in experimental or control group in the certain period of time and that is the reason for having the different numbers of participants in two groups.